# ^1^H-NMR Metabolomics as a Tool for Winemaking Monitoring

**DOI:** 10.3390/molecules26226771

**Published:** 2021-11-09

**Authors:** Inès Le Mao, Jean Martin-Pernier, Charlyne Bautista, Soizic Lacampagne, Tristan Richard, Gregory Da Costa

**Affiliations:** University of Bordeaux, INRAE, Bordeaux INP, UR OENO, EA 4577, USC 1366, F-33140 Villenave d’Ornon, France; ines.le-mao@u-bordeaux.fr (I.L.M.); jean.martin@u-bordeaux.fr (J.M.-P.); charlynebautista@hotmail.com (C.B.); soizic.lacampagne@u-bordeaux.fr (S.L.); gregory.da-costa@u-bordeaux.fr (G.D.C.)

**Keywords:** ^1^H-NMR, metabolomics, wine, winemaking

## Abstract

The chemical composition of wine is known to be influenced by multiple factors including some viticulture practices and winemaking processes. ^1^H-NMR metabolomics has been successfully applied to the study of wine authenticity. In the present study, ^1^H-NMR metabolomics in combination with multivariate analysis was applied to investigate the effects of grape maturity and enzyme and fining treatments on Cabernet Sauvignon wines. A total of forty wine metabolites were quantified. Three different stages of maturity were studied (under-maturity, maturity and over-maturity). Enzyme treatments were carried out using two pectolytic enzymes (E1 and E2). Finally, two proteinaceous fining treatments were compared (vegetable protein, fining F1; pea protein and PVPP, fining F2). The results show a clear difference between the three stages of maturity, with an impact on different classes of metabolites including amino acids, organic acids, sugars, phenolic compounds, alcohols and esters. A clear separation between enzymes E1 and E2 was observed. Both fining agents had a significant effect on metabolite concentrations. The results demonstrate that ^1^H-NMR metabolomics provides a fast and robust approach to study the effect of winemaking processes on wine metabolites. These results support the interest to pursue the development of ^1^H-NMR metabolomics to investigate the effects of winemaking on wine quality.

## 1. Introduction

Over the last decade, the number of studies on wine has grown steadily as quality and authenticity have become major concerns for both producers and consumers. Thus, fine wine analysis has blossomed to study and characterize each step of the production chain from the vineyard to the finished wine. Metabolomics has emerged from the studies that have flourished, which studies all the small molecules of a biological system. Metabolomics science, by adopting a targeted or non-targeted approach, can be discriminating, predictive and informative depending on the subject of the study [1,2].

A well-known challenge in the study of wine is the complexity of the matrix. Indeed, hundreds of compounds of different chemical families make analysis complex. The two most commonly used analytical techniques for wine metabolomics are mass spectrometry and NMR [3]. Although MS is a more sensitive tool, NMR combines many advantages that make it a tool of choice for metabolomic analysis of wines [4,5,6,7]: no complex sample preparation, low volume requirements, fast and reproducible analysis and detection of a large number of compounds belonging to different chemical families, such as organic acids, alcohols, sugars, phenolic compounds and amino acids [8,9]. Moreover, NMR is a wine screening tool that can also provide quantitative data thanks to [10].

Since the chemical composition of wine is known to be influenced by multiple factors, many wine NMR studies have been focused on the impact of grape variety [11,12], geographical origin [9,13,14], vintage and aging [15,16]. Comparatively, few NMR technique-based studies have been carried out on viticultural practices and winemaking processes, which can also influence the metabolite profile of wine, and thus its organoleptic properties [17,18]. Fermentation processing was explored by establishing the difference between alcoholic and malolactic fermentation processes on metabolites with a time course evolution [19], while other studies focused on the impact of using different yeast strains during white wine fermentation [20], and red wine fermentation [11]. Lee and collaborators established, by a combination of ^1^H-NMR and gas chromatography approaches, the fermentative behaviors and the metabolic variations of five bacteria strains during malolactic fermentation in Meoru wines, in order to reduce wine acidity [15]. Concerning winemaking technologies, through metabolite profiles, De Pascali and collaborators analyzed the difference between winemaking processes (ultrasound, cryomaceration and traditional processes) on Negroamaro red wine [21], while Baiano and collaborators examined the difference between traditional, cryomaceration and reductive processes and a combination of the last two procedures on Sauvignon Blanc wine [22].

Wines are subjected to different winemaking processes including the addition of different agents for better wine stability and limpidity, such as finings, used to obtain clarified and limpid wines [23], and enzymatic treatments, used to enhance the degradation of grape berry cell walls and the extraction of aroma and phenolic compounds [24]. The control of these different processes is a crucial point in order to produce quality wines. The impact of finings on wine composition has been studied using different techniques including spectrophotometry or liquid and gas chromatography [23,25]. In the same way, different techniques have been used to study the effect of enzymatic addition on the release of phenols, monosaccharides and polysaccharides, by different chromatographic techniques [24,26].

The aim of this study was to determine the ability of ^1^H-NMR spectroscopy combined with multivariate statistical analyses to discriminate different winemaking parameters and processes (maturity, fining and enzymatic addition) and to explore the impact of these processes on wine composition.

## 2. Results and Discussion

### 2.1. ^1^H-NMR Analysis of Wine

The typical ^1^H-NMR spectra of wine after solvent suppression are presented in Figure 1. The signals at 0.00 and 8.28 ppm correspond to TMSP and calcium formate, respectively. All other signals correspond to wine constituents. Based on previous studies [8,17], 2D NMR analyses and pure chemical standards, forty-eight constituents were identified (Table 1), including amino acids, organic acids, alcohols, carbohydrates, phenolic compounds, esters, ketones and aldehydes.

### 2.2. Maturity Stages

The ability of ^1^H-NMR-based metabolomics to discriminate wines produced from grapes harvested at three different stages of maturity (M1: under-maturity; M2: maturity; M3: over-maturity) was assessed using multivariate analysis. Principal component analysis (PCA) followed by orthogonal partial least square discriminant analysis (OPLS-DA) was performed on the quantification data extracted from the ^1^H-NMR spectra. The analysis of the maturity effect was carried out using six replicates for each maturity stage.

The PCA score plot (R^2^X(cum) and Q^2^(cum) of 0.480 and 0.130, respectively) showed separation between the three stages of maturity (Figure 2a). To confirm this observation, OPLS-DA (number of latent variables 2 + 2 + 0) was performed (Figure 2b). The OPLS-DA score plots confirmed discrimination between wines with high validated predictability (Q^2^ = 0.873) and goodness of fit values (R^2^X and R^2^Y of 0.615 and 0.975, respectively). The validity of the OPLS-DA model was confirmed by cross-validated ANOVA (CV-ANOVA) providing a significant *p*-value (3.12 × 10^−5^). The recognition ability of the model was 100%, as shown in the misclassification table (Appendix A). The under-maturity, maturity and over-maturity stages were completely distinguished from one another. This first result indicates that ^1^H-NMR metabolomics is able to discriminate wines made from berries collected at different stages of maturity. To analyze the effect of under-mature and over-mature grapes on wine quality, the OPLS-DA model was further applied to discriminate the differential constituents between these wines and those made with grapes harvested at maturity. The OPLS-DA score and loading plots are shown in Figure 2c,d, respectively, for comparison of under-maturity/maturity (M1/M2), and in Figure 2e,f, respectively, for comparison of over-maturity/maturity (M3/M2). The OPLS-DA model parameters (Appendix A) and cross-validation procedure demonstrated an excellent modeling and predictive ability in each case.

To identify the contribution of wine constituents to the discrimination between different groups, a combination of variable importance in projection (VIP) and p(corr) values was used [27]. Potential differential compounds were selected with absolute VIP and p(corr) values greater than 1.0 and 0.5, respectively. The highlighted wine constituents were further controlled by ANOVA followed by Tukey’s multiple comparison test, and only those with a *p*-value of <0.05 were conserved. A total of 16 wine constituents including amino acids (alanine, histidine, leucine and tyrosine), organic acids (citric, lactic, tartaric and shikimic acids), sugar derivatives (glucose, galacturonic acids and xylose), phenolic compounds (catechin, epicatechin and gallic acid), an alcohol (methanol) and an ester (ethyl lactate) had a strong influence in distinguishing M2 from M1 wines. Wines made with grapes harvested at under-maturity contained less alanine, histidine, leucine, tyrosine, citric acid, tartaric acid, glucose, galacturonic acid, gallic acid, ethyl lactate and methanol (Appendix A). On the contrary, they contained more lactic acid, shikimic acid, xylose, catechin and epicatechin than wines made with grapes harvested at maturity. Similarly, a total of 10 wine constituents including organic acids (citric acid, lactic acid and shikimic acid), sugar derivatives (glucose, galacturonic acids and xylose) and alcohols (acetoin, 2,3-butanediol, glycerol and methanol) had a strong influence in separating M3 and M2. Wines made with grapes harvested at over-maturity contained more citric acid, glucose, galacturonic acid, 2,3-butanediol, glycerol and methanol (Appendix A). On the contrary, they contained less lactic acid, shikimic acid, xylose and acetoin than wines made with grapes harvested at maturity.

Most studies explored the ripening effect on the berries’ metabolome [28], but few focused on the wine metabolome. Nevertheless, grape maturity generated significant sensory effects on wines [29]. In this study, a significant effect of grape maturity was observed on the Cabernet Sauvignon wine metabolome using ^1^H-NMR metabolomics.

The overall trends observed are in agreement with the results obtained by Bindon et al. on Cabernet Sauvignon [30]. Even though the ethanol concentration increased from 13.7 at M1 to 14.2% at M3, this variation was not significant. Nevertheless, a significant increase in other alcohols such glycerol was observed, particularly between M2 and M3 (Appendix A). As expected, the higher lactic acid concentrations in M1 (mean 1.90 ± 0.09 g/L) samples can be explained by the higher initial malic acid concentrations in grapes harvested at under-ripeness (Appendix A). Indeed, as the finished wines underwent malolactic fermentation, the initial malate was partly transformed into lactate by lactic acid bacteria [31]. On the contrary, a significant increase in tartaric acid was observed between M1 and M2 samples. This result could be associated with the increase in tartaric acid in grapes between under-maturity and maturity (Appendix A). Regarding the other acids, no significant variation was observed except for citric acid and shikimic acid. This observation is in agreement with literature data [30]. Practically all reducing sugars (glucose and fructose) were consumed, indicating the efficiency of fermentation. A higher content was observed in M3 samples (mean 0.89 ± 0.15 g/L for glucose and fructose). As previously reported [32], proline was the main amino acid observed in wines (mean range 0.801–0.962 g/L), and the highest concentration was measured in M3 samples but with no significant difference from other modalities. In addition, a significant increase in four amino acids was observed between M1 and M2 samples (Appendix A). Interestingly, some specific trends were observed such as the significant variation in acetoin between M2 and M3 samples (means 23 ± 2 and 11 ± 3 mg/L, respectively). This compound is a well-known precursor of the biosynthesis of 2,3-butanediol, which contributes to the bouquet of the wine. To conclude, ^1^H-NMR metabolomics was able to discriminate wines produced from grapes at different stages of ripeness, providing informative trends on the potential quality of the wine.

### 2.3. Enzyme Treatments

Enzymes are used to facilitate hydrolysis of the cell walls, providing a more efficient extraction of phenolic compounds [33]. The effect of two different preparations, E1 and E2, was monitored using ^1^H-NMR-based metabolomics and multivariate analysis. Both preparations are pectolytic enzymes. Five replicates for each enzyme were analyzed.

Wine samples were treated by enzymes E1 and E2. The data extracted from ^1^H-NMR spectra were compared by PCA and OPLS-DA. The PCA score plot exhibited separation between the three modalities (Appendix A). To better emphasize the discrimination, OPLS-DA was performed. Score and loading plots are shown in Figure 3a,b, respectively. A clear tendency to discriminate the enzyme treatments was observed (Q^2^, R^2^X and R^2^Y of 0.617, 0.869 and 0.518, respectively). To validate the model, a cross-validation procedure was performed, which indicated that all wines were correctly classified (Appendix A). ^1^H NMR-based metabolomics allows observing the effect of the use of enzymes independent of the grape maturity stage. To analyze the effect of each enzymatic preparation, the OPLS-DA model was further applied to discriminate the differential constituents between these wines and untreated samples. Discriminating constituents were identified based on VIP and absolute p(corr) values and confirmed by ANOVA followed by Tukey’s multiple comparison test. A total of 11 compounds were identified (Figure 3c).

Wine samples treated by enzymes contained significantly more arabinose, galacturonic acid, glucose and catechin and less xylose, myo-inositol and succinic acid. As expected, the enzyme treatments had a significant effect on wine polysaccharide composition [24]. Pectolytic enzymes reduce the molecular weight of polysaccharides, resulting in an increase in free monosaccharides. Significant increases in galacturonic acid, arabinose and glucose have previously been reported in wine samples treated by enzyme preparations [34]. These are clear indicators of the pectolytic activity of enzymes resulting from the degradation of pectic polysaccharides in the cell walls of the grape berry. The increase in galacturonic acid, well above the sensory detection threshold (125 mg/L [35]), could contribute to the wine sourness. Surprisingly, the levels of xylose decreased in both enzymatic treatments. A similar effect was observed in Merlot red wines after enzyme treatments [36]. This could be due to a competitive effect between the used enzymes and endogenous pectinases that are naturally present in the grape solids. The specific significant increase in catechin confirms the ability of enzymes to enhance phenolic extraction during winemaking [36]. Succinic acid is the main nonvolatile acid produced by yeasts during alcoholic fermentation [37]. Even though this compound could be formed by yeasts until the stationary phase of alcoholic fermentation, the decrease in succinic acid observed with both enzymes cannot be explained by the modulation of yeast activity. An interaction with the products released by the enzymes could be at the origin of this decrease. Interestingly, specific trends of enzymatic preparation E2 were observed (Figure 3c). The use of preparation E2 induced a higher level of arabinose than that from E1. The release of significant quantities of arabinose could be due to an extensive and specific degradation of arabinose-rich polysaccharides by E2. On the contrary, E2 significantly reduced the production of acetoin, isobutanol, acetic acid and leucine in comparison to the E1 and control modalities. As previously mentioned, acetoin is the precursor of 2,3-butanediol, and isobutanol is a higher alcohol involved in wine aroma perception [38]. These results demonstrate the ability of ^1^H-NMR metabolomics to monitor the effects of enzyme treatment in winemaking processes.

### 2.4. Fining Treatments

Fining treatments are mainly used in winemaking to clarify and stabilize the wine [39]. They bind to specific compounds (proteins, polymeric phenols or tannins) and form insoluble complexes that can be removed from the wine. Depending on the desired objective, different classes of fining agents can be used. Due to their capacity to bind with tannins, fining agents having a proteinaceous origin are widely used. In this study, the effects of two different fining agents on the wine metabolome were studied by using ^1^H-NMR metabolomics. Two fining agents of proteinaceous origin were investigated: a vegetable protein (F1), and a mix of pea protein and PVPP (F2).

Wine samples were treated by finings F1 and F2. The data extracted from ^1^H-NMR spectra were compared by PCA (Appendix A) followed by OPLS-DA to confirm the observed trends. OPLS-DA score and loading plots are shown in Figure 4a,b, respectively. A clear discrimination between the three sets of wine was observed in the OPLS-DA score plot (Q^2^, R^2^X and R^2^Y of 0.666, 0.897 and 0.455, respectively). To validate the model, a cross-validation procedure was performed, which indicated that all wines were correctly classified (Appendix A). As a first result, these data indicate that ^1^H-NMR metabolomics is able to monitor the impact of proteinaceous finings on the wine metabolome. These treatments can induce precipitation of phenolic compounds, particularly polymerized and galloyled tannins [40], affecting the wine organoleptic properties. As shown in Figure 4C, the wines treated by fining F1 presented a higher concentration of amino acids than the untreated wines. These wines contained significantly higher concentrations of alanine and threonine (51 ± 4 and 45 ± 5 mg/L, respectively) in addition to showing an increase in choline (mean 33 ± 3 mg/L) in comparison to the untreated wines and fining F2 treatment. It is surprising to observe an increase in some constituents under the action of the finings. These agents are rather known to precipitate wine constituents. Nevertheless, Maury et al. observed a significant increase in amino acids due to proteinaceous fining treatment in a model wine solution [41]. In addition, a specific action of each fining could be observed. While fining F1 had no significant impact on the main constituents observed by ^1^H-NMR, fining F2 induced a significant decrease in some wine constituents such as trigonelline, phenethyl alcohol and isoamyl alcohol (Figure 4c). Phenethyl alcohol is the most important phenolic higher alcohol present in wine [42]. A recent study showed that certain processes can reduce the content of phenyl alcohol in addition to other volatile compounds [43]. Isoamyl alcohol, together with phenethyl alcohol, isobutanol and methionol, is considered one of the most aroma-powerful higher alcohols [38]. In addition, isoamyl alcohol seems to reduce fruity and woody notes in wines. Even though higher alcohol effects depend on the wine aromatic context, ^1^H-NMR metabolomics approaches could provide valuable data concerning the effects of fining treatments on wine organic properties.

## 3. Materials and Methods

### 3.1. Wine Samples

For this study, Cabernet Sauvignon grapes were harvested in 2019 from a vineyard of the Bordeaux Superieur wine region. The grapes were harvested at three different stages of maturity: under-maturity (M1), maturity (M2) and over-maturity (M3), one week between each harvest was observed. Maturity was monitored using a set of different parameters including pH, sugar content, total acidity and malic acid level. These parameters were measured following OIV methods (OIV-OENO 598-2018, OIV-OENO 599-2018, OIV-OENO 600-2018, OIV-MA-AS311-02, OIV-MA-F1-06, OIV-MA-F1-09, OIV-MA-AS313-01) using an Analyzer Y15 (BioSystems, Barcelona, Spain). Data are reported in Appendix A.

### 3.2. Winemaking

Wines were elaborated at the nanovinification platform (Bordeaux Vinif) of the Institute of Vine and Wine Sciences of Bordeaux (ISVV) following a specific nanovinification method [44]. Briefly, grapes from each of the 3 modalities were destemmed and crushed. The crushed grapes were macerated in 10 L stainless-steel tanks. At the same time, bisulfite (5 mg/L of must) was added. Fermentations were initiated with Actiflore F33 (commercial dry Saccharomyces cerevisiae yeast, Laffort, France) inoculated at 0.1 g/L. Each lot was fermented to completion. The process was monitored by density measurement using the electronic densimeter DMA 35 Basic (Anton Paar France, Ulis, France). At the end of alcoholic fermentation, each lot was pressed at 1 bar and transferred to 5 L glass tanks. Malolactic fermentations were performed after direct inoculation with commercial Oenoccocus oeni bacteria SB3 Direct (Laffort, Bordeaux, France). Three weeks after the beginning of winemaking, the wines were racked, and sulfur dioxide was adjusted to 50 mg/L of wine. The wines were cold stabilized at 4 °C for 1 week, bottled and stored in the experimental wine cellar at 15 °C until treatment and analysis. The effects of two enzymes and two fining agents were compared to untreated samples. For enzyme treatments, two different pectolytic enzymes were used at the concentration recommended by the provider: enzyme 1 (E1) at 4 g/hL, and enzyme 2 (E2) at 3 mL/hL. The preparations were added at the end of malolactic fermentation. Samples were kept at 15 °C for 15 days before analysis. For fining processes, fining agents were added, after malolactic fermentation, to 50 mL of wine respecting the proportions commonly used in wineries: fining 1 (F1), vegetable protein at 3 g/hL; fining 2 (F2), pea protein and PVPP (polyvinylpolypyrrolidone) at 80 g/hL. Samples were kept at 15 °C for 3 days before analysis. The supernatants were collected for the preparation of the NMR analyses. At least five biological replicates were analyzed for each treatment.

### 3.3. Sample Preparation

Wines samples were prepared according to our previous work [9]. Briefly, 420 μL wine was mixed with 120 μL phosphate buffer (1 M, pH 2.6), and 60 μL deuterated water containing 0.5 mM sodium trimethylsilylpropanoic acid (TMSP) and 7 mM calcium formate. The pH was adjusted to 3.1 using a small-scale semi-automatic system (BTpH, Bruker BioSpin, Rheinstetten, Germany) using a 1 M HCl solution.

### 3.4. NMR Analysis

The ^1^H-NMR spectra were recorded on a 600 Mhz Avance III NMR spectrometer (Bruker, Wissembourg, France) operating at 600.27 MHz, equipped with a TXI 5 mm probe with z gradient coils. The measurements were performed at 293 K, using Topspin 4.0.8 software (Bruker, Wissembourg, France). Three different sequences were used: the ZG30 and ZGPR sequences were set at 8 ns (number of scans) and NOESYGPPS1D was set at 32 ns for water and ethanol suppressions.

The acquisition parameters were set as follows: The free induction decay (FID) was collected into a time domain (TD) of 64K data points, with a spectral width (SW) of 16 ppm, an acquisition time (AQ) of 3.40 s and a relaxation delay (RD) of 5 s per scan. Then, 90° pulse calibration was carried out for each sample automatically, and the shimming was set manually in gs mode for each spectrum in order to obtain the finest possible line width (lower than 1 Hz). Water and ethanol signal suppressions were achieved during RD using a shaped pulse with a multiple-band selective solvent suppression (7 Hz centered on each ethanol signal, and 20 Hz for water), with a power level for presaturation of 50.37 dB and a shaped pulse for presaturation of 34.83 dB.

The FID was multiplied by an exponential function corresponding to a 0.3 Hz line-broadening factor prior to the Fourier transform. Manual phase and baseline correction was applied to the resulting spectrum, which was then manually phased and aligned to zero using the TMSP signal. Wine constituents were identified based on previous studies [8,17], 2D NMR analyses and pure chemical standards. Compounds were quantified by targeted analysis, by the global spectral deconvolution method (GSD) [45], using the simple mixture analysis (SMA) plugin of MestReNova 12.0 software (Mestrelab Research, Santiago de Compostela, Spain). Calcium formate was used as an internal standard for quantitation.

### 3.5. Multivariate Data Analysis

Statistical analyses were carried out using SIMCA 16.0 software (Sartorius, Goettingen, Germany). No data preprocessing was performed prior to statistical analysis. Principal component analysis (PCA) was performed to visualize and observe the distribution of the metabolic variance of the dataset. Orthogonal partial least square discriminant analysis (OPLS-DA) was performed to sharpen the separation between observation groups [27,46]. OPLS-DA was developed as an improvement of the PLS-DA procedure to discriminate classes [47,48]. The results were visualized in the form of score plots and loading plots. Hotelling’s T^2^ ellipse, in score plots, defines the 95% confidence region. The model quality was described by R-squared (R^2^) and Q-squared (Q^2^) values. R^2^X and R^2^Y represent the goodness of fit or explained variation, and Q^2^ represents the goodness of prediction or predicted variation [49]. Results are presented in the form of score plots (each point shows an individual wine sample) and loading plots (each coordinate shows one wine constituent). The variable importance of projection (VIP) represents the total importance of each variable in explaining the model. The p(corr) value represents the correlation coefficient between the model and original data (ranged between –1.0 and 1.0) for each loading. OPLS-DA models were validated by analysis of variance testing of cross-validated predictive residuals (CV-ANOVA), with the p-value indicating the model significance. The recognition abilities of the OPLS-DA models were assessed using misclassification tables. The potential discriminant metabolites were identified using VIP and p(corr) values [27]. Metabolites with VIP and absolute p(corr) values larger than 1.0 and 0.5, respectively, were considered potential discriminant metabolites. One-way analysis of variance (ANOVA) followed by Tukey’s multiple comparison procedure was used to identify the discriminant metabolites.

## 4. Conclusions

In our previous studies, we showed that NMR is a valuable tool for ensuring wine traceability complementary to the OIV’s official methods [9,50]. NMR combines several advantages that make it a tool of choice for wine metabolomic analysis. In this study, our main goal was to elucidate whether NMR metabolomics could provide valuable data to monitor the effects of winemaking processes including grape maturation and enzyme and fining treatments. The results indicate that NMR metabolomics can discriminate these different winemaking processes. This technique is efficient in quantifying different wine constituents and has the advantages of being fast and reproducible. Combined with multivariate statistical analyses, it allows the identification of some impacted metabolites by each specific winemaking practice. As a complement to other analytical techniques and sensory evaluations, NMR metabolomics could provide valuable new data concerning the impact of winemaking practices on wine organoleptic properties. Additionally, more research is needed to better understand the potential contribution of NMR metabolomics to the investigation of the effects of winemaking practices on wine quality.

## Figures and Tables

**Figure 1 molecules-26-06771-f001:**
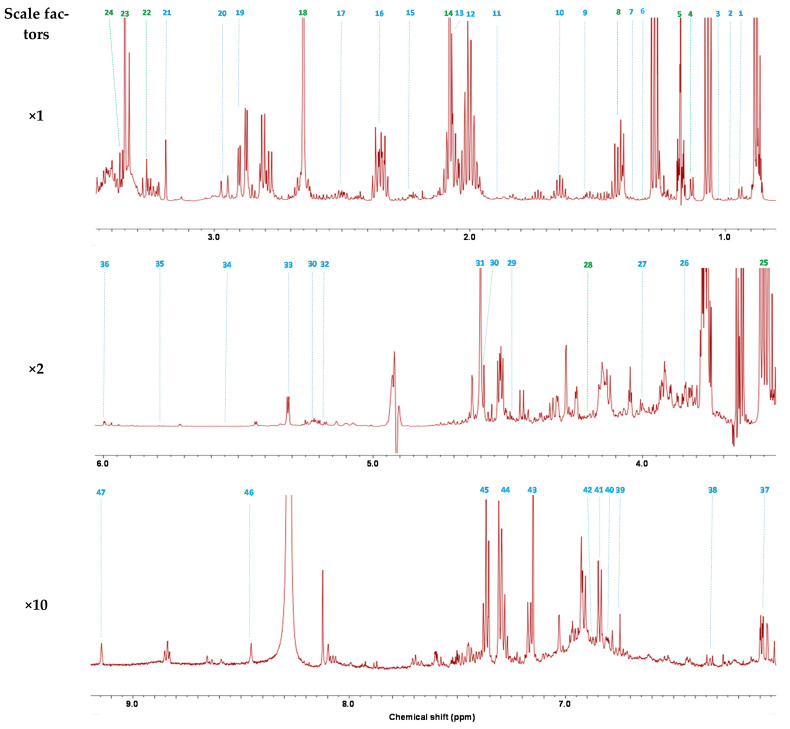
Typical ^1^H-NMR spectrum of wine after water and ethanol suppression (NOESYGPPS1D sequence). Identified constituents are listed in Table 1 (compounds in green are quantified on the ZGPR sequence).

**Figure 2 molecules-26-06771-f002:**
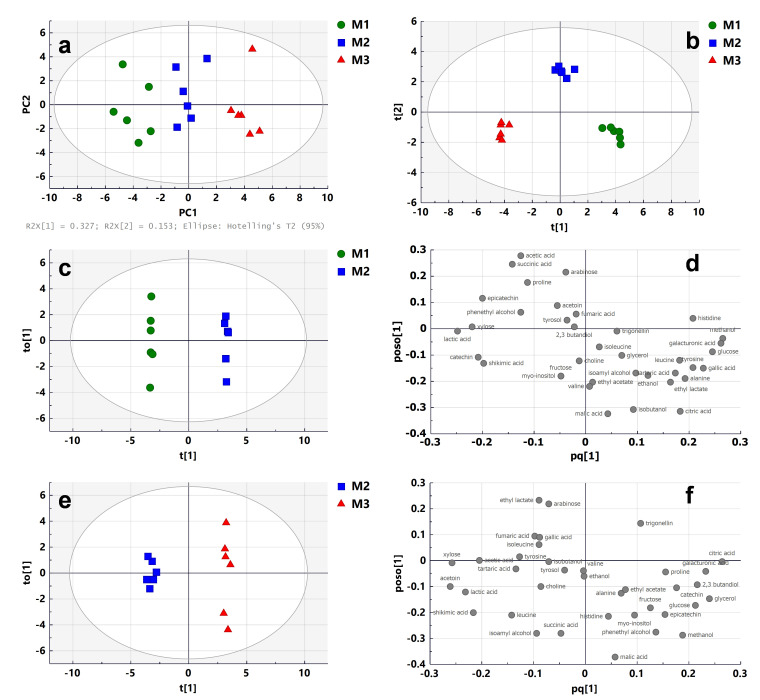
Multivariate analysis of ^1^H-NMR spectra of wine samples from grapes harvested at three different stages of maturity (M1: under-maturity; M2: maturity; M3: over-maturity): (**a**) PCA score plot; (**b**) OPLS-DA score plot; (**c**) OPLS-DA score showing separation of M1 and M2 samples; (**d**) loadings from OPLS-DA between M1 and M2 samples; (**e**) OPLS-DA score showing separation of M2 and M3 samples; (**f**) loadings from OPLS-DA between M2 and M3 samples (t[1] and to[1]: first predictive and orthogonal components; t[2]: second predictive component; pq[1] and poso[1]: predictive and orthogonal component loadings).

**Figure 3 molecules-26-06771-f003:**
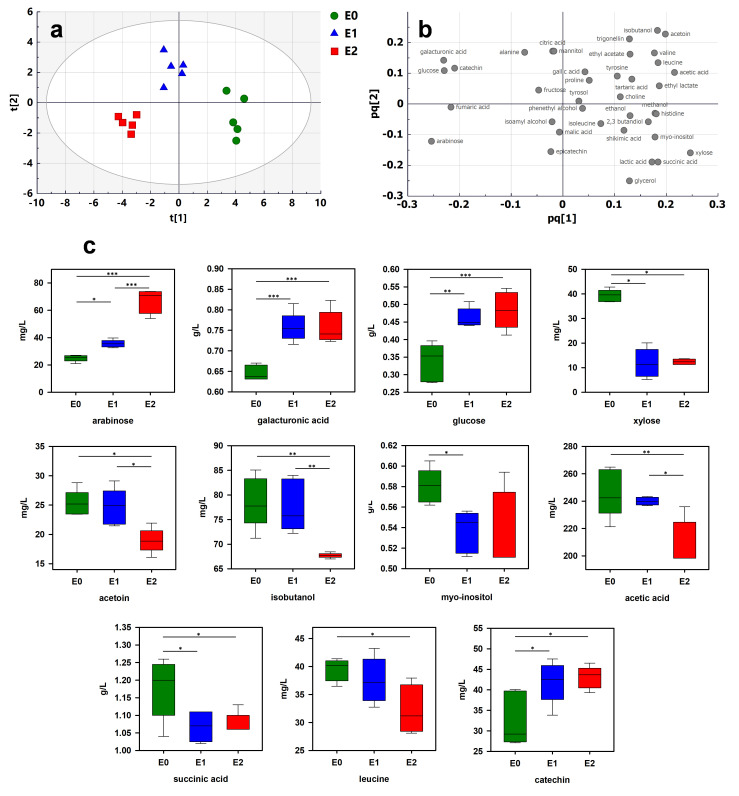
Multivariate analysis of ^1^H-NMR spectra of wine samples treated by different enzymes (E0: untreated; E1: enzyme 1; E2: enzyme 2): (**a**) OPLS-DA score plot; (**b**) loading plot; (**c**) boxplots of 11 most discriminant wine constituents (t[1] and t[2]: first and second predictive components; pq[1] and pq[2]: first and second predictive component loadings). The significance in the difference was calculated by ANOVA followed by Tukey’s multiple comparison test (indicated as * *p* < 0.05, ** *p* < 0.01, *** *p* < 0.001).

**Figure 4 molecules-26-06771-f004:**
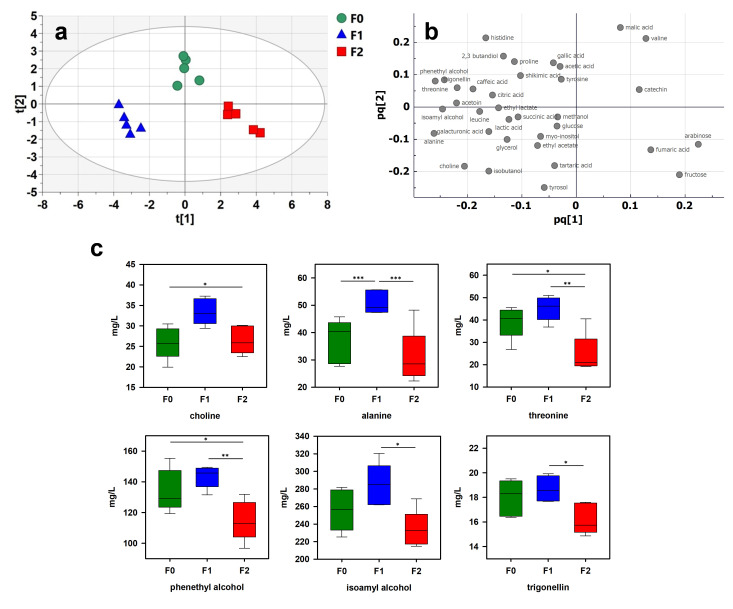
Multivariate analysis of ^1^H-NMR spectra of wine samples treated by different finings (F0: untreated; F1: fining 1; F2: fining 2): (**a**) OPLS-DA score plot; (**b**) loading plot; (**c**) boxplots of 6 most discriminant wine constituents (t[1] and t[2]: first and second predictive components; pq[1] and pq[2]: first and second predictive component loadings). The significance in the difference was calculated by ANOVA followed by Tukey’s multiple comparison test (indicated as * *p* < 0.05, ** *p* < 0.01, *** *p* < 0.001).

**Table 1 molecules-26-06771-t001:** Chemical shifts and coupling constants used for compound identification. The signals chosen for quantitation are in bold.

	Compound	δ^1^_H_ (Multiplicity, *J* in Hz, Assignment)
1	leucine	**0.96** (*d*, 6.2, 2CH_3_), 1.71 (*m,* CHCH_2_), 3.74 (*m*, CH)
2	isoleucine	0.93 (*t*, 7.4, CH_3_), **0.99** (*d*, 7.0, CH_3_), 1.24 (*m*, CH_2_), 1.45 (*m*, CH_2_), 1.97 (*m*, CH), 3.66 (*d*, 3.9, CH)
3	valine	**0.99** (*d*, 7.3, CH_3_), **1.04** (*d*, 7.3, CH_3_), 2.28 (*m*, CH), 3.66 (*d*, 4.3, CH)
4	2,3-butanediol	**1.13** (*d*, 6.2, 2CH_3_), 3.61 (*m*, 2CH)
5	ethanol	**1.17** (*t*, 7.2, CH_3_), 3.65 (*q*, CH_2_)
6	threonine	**1.32** (*d*, 6.7, CH_3_), 2.58 (*d*, 4.9, CH), 4.24 (*m*, CH)
7	acetoin	**1.37** (*d*, 7.0, CH_3_), 2.21 (*s*, CH_3_), 4.42 (*q*, CH)
8	lactic acid	**1.40** (*d*, 7.0, CH_3_), 4.31 (*q*, 7.0, CH)
9	alanine	**1.50** (*d*, 7.2, CH_3_), 3.76 (*q*, CH)
10	isopentanol	0.88 (*d*, 6.7, 2CH_3_), 1.44 (*q*, CH_2_); **1.66** (*m*, CH), 3.61 (*t*, 6.7, CH_2_)
11	arginine	1.70 (*m*, CH_2_), **1.89** (*m*, CH_2_), 3.23 (*t*, CH_2_), 3.75 (*t*, 6.5, CH)
12	proline	**1.99** (*m*, CH_2_), 2.06 (*m*, CH), 2.33 (*m*, CH), 3.32 (*dt*, 14.0, 7.1, CH), 3.42 (*dt*, 11.6 and 7.0, CH), 4.11 (*dd*, 8.6 and 6.4, CH)
13	ethyl acetate	1.26 (*t*, 7.2, CH_3_), 4.12 (*q*, CH_2_), **2.07** (*s*, CH_3_)
14	acetic acid	**2.08** (*s*, CH_3_)
15	ethanal	**2.23** (*d*, 3.0, CH_3_), 9.67 (*q*, CH)
16	pyruvic acid	**2.35** (*s*, CH_3_)
17	γ-aminobutyric acid	1.96 (*m*, CH_2_), **2.50** (*t*, 7.3, CH_2_), 3.05 (*m*, CH_2_)
18	succinic acid	**2.65** (*s*, 2CH_2_)
19	malic acid	2.78 (*dd*, 16.3 and 7.0, CH), **2.89** (*dd*, 16.3 and 4.5, CH), 4.53 *(dd*, CH)
20	citric acid	2.79 (*d*, 15.6, CH_2_), **2.94** (*d*, 15.6, CH_2_)
21	choline	**3.19** (*s*, 3CH_3_), 3.51 (*dd,* CH_2_), 4.05 (*m*, CH_2_)
22	myo-inositol	**3.27** (*t*, 9.7, CH), 3.52 (*dd*, 10.0 and 2.8, 2CH), 3.61 (*t*, 2.8, 2CH), 4.05 (*t*, 2.8, CH)
23	methanol	**3.35** (*s*, CH_3_)
24	isobutanol	0.87 (*d*, 6.7, 2CH_3_), 1,73 (*m*, CH), **3.36** (*d*, 6.7, CH_2_)
25	glycerol	**3.55** (*dd*, 11.8 and 6.5, CH_2_), 3.64 (*dd*, CH_2_), 3.77 (*m*, CH)
26	mannitol	3.65 (dd, 11.7, 6.2 CH_2_), 3.73 (*m*, CH), 3.77 (*d*, 9.0, CH), **3.84** (*dd*, 11.9, 2.8, CH_2_)
27	fructose	3.56 (*m*, CH_2_), 3.70 (*m*, 2CH_2_), 3.77 (*m*, CHCHCH_2_), 3.87 (*dd,* 9.9, 3.4, CH), 3.97 (*m*, CH), **4.00** (*dd*, 12.8, 1.0 CH_2_), 4.09 (*m,* 2CH)
28	ethyl lactate	1.28 (*t*, CH_3_), **1.42** (*d*, 7.0, CH_3_), 4.22 (*q*, 7.06, CH), 4.39 (*q*, 7.0, CH)
29	arabinose	3.51 (*dd*, CH), 3.68 (*m*, CHCH_2_), 3.83 (*dd*, CH), 3.90 (*m*, CHCH_2_), 3.95(*m*, CH), 4.02 (*m*, CHCH_2_), **4.50** (*d*, 7.7, CH), 5.25 (*d*, CH)
30	glucose	3.23 (*dd,* 9.2, 8.0, CH), 3.39 (*m*, CH), 3.45 (*dd*, 9.8, 3.7, CH), 3.72 (m, CHCH_2_), 3.82 (*m,* CHCH_2_), 3.88 (*dd,* 12.2, 2.1, CH_2_), 4.63 (*d*, 7.9, CH), **5.22** (*d*, 3.6, CH)
31	tartaric acid	**4.60** (*s*, 2CH)
32	xylose	3.21 (*dd*, 9.3, 7.9, CH), 3.31 (*t*, 11.4, CH_2_), 3.42 (*t*, 9.25, CH), 3.51 (*dd*, 9.3, 3.7, CH), 3.63 (*m*, CHCHCH_2_), 3.91 (*dd*, 11.5, 5.5, CH_2_), 4.57 (*d*, 7.9, CH), **5.19** (*d*, 3.7, CH)
33	galacturonic acid	3.49 (*dd*, 8.0, 10.0, CH), 3.69 (*dd*, 9, 3.5, CH), 3.80 (*dd*, 10.3, 3.8, CH), 3,92 (*dd*, 10.3, 3.4, CH), 4.24 (*dd*, 3.6, 1.2, CH), 4.26 (*d*, 1.2, CH), 4.31 (*dd*, 3.3, 1.4, CH), 5.32 (*d*, 3.8, CH)
34	glucuronic acid	3.29 (*t*, 8.6, CH), 3.51 (*m*, 2CH), 3.58 (*dd*, 9.7, 3.7, CH), 3.73 (*m*, 2CH), 4.08 (*d*, 10.8, CH), 4.64 (*d*, 7.9, CH), 5.25 (*d*, 3.7, CH), **5.55** (*d*, 4, CH)
35	sorbic acid	1.82 (*d*, 6.2, CH_3_), **5.78** (*d*, 15.3, CH), 6.25 (*m,* 2CH), 7.16 (*dd*, 15.3, 10.3, CH)
36	epicatechin	2.76 (*m*, CH_2_), 2.90 (*m*, CH_2_), 4.32 (*m*, CH), 4.95 (*m*, CH), 6.09 (*d*, 2.0, CH), **6.12** (*d*, 2.0, CH), 6.93 (*m*, CH_2_), 7.03 (*d*, 2.0, CH)
37	catechin	2.53 (*dd*, CH_2_), 2.85 (*m*, CH_2_), 4.15 (*m*, CH), 4.41 (*d*, 7.0, CH), **5.99** (*d*, 2.0, CH), 6.08 (*d*, 2.3, CH), 6.84 (*d*, 8.6, CH), 6.92 (*m*, 2CH)
38	caffeic acid	**6.33** (*d*, 16.0, CH), 6.92 (*d*, 8.0, CH), 7.07 (*dd*, 8.2, 2.0, CH), 7.14 (*d*, 2.0, CH), 7.29 (*d*, CH)
39	fumaric acid	**6.78** (*s*, 2CH)
40	shikimic acid	2.21 (*dd*, 18.2,7.0, CH_2_), 2.75 (*dd*, 18.0, 5.3, CH_2_), 3.74 (*dd*, 8.6, 4.3, CH), 4.01 (*m*, CH), 4.42 (*t*, 4.1, CH), **6.82** (*dt*, CH)
41	tyrosol	2.77 (*t,* CH_2_), 3.77 (*t*, CH_2_), **6.84** (*m*, 8.4, 2CH), 7.17 (*m*, 8.4, 2CH)
42	tyrosine	3.02 (*dd*, CH_2_), 3.17 (*dd*, CH_2_), 3.92 (*dd*, CH), **6.86** (*m*, 8.4, 2CH), 7.17 (*m*, 8.6, 2CH)
43	gallic acid	**7.16** (*s*, 2CH)
44	phenethyl alcohol	2.85 (*t*, 6.62, CH_2_), 3.74 (*t*, CH_2_), **7.33** (*m*, 5CH)
45	syringic acid	3.84 (*s*, 2CH_3_), 7.36 (s, 2CH)
46	histidine	3.16 (*dd*, 15.6, 7.7, CH), 3.23 (*dd*, 16.0, 5.0, CH), 3.98 (*dd*, 7.7, 5.0, CH), 7.09 (*d*, 5.0, CH), 7.90 (*d*, 1.1, CH)
47	trigonelline	4.42 (*s*, CH_3_), 8.07 (*m*, CH), 8.82 (*m*, 2CH), 9.11 (*s*, CH)

## Data Availability

Data are contained within the article, and they are available from the first corresponding author.

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
