# Peer review of "^1^H-NMR Metabolomics as a Tool for Winemaking Monitoring"

_molecules, 2021, doi:10.3390/molecules26226771_

Round 1

Reviewer 1 Report

The reviewed paper (Manuscript ID molecules-1440327) examined the ability of 1H-NMR spectroscopy combined with multivariate statistical analysis for the Cabernet Sauvignon wine discrimination. The influence of winemaking parameters and processes, including grape maturity, enzyme treatment, and fining process, on the wine constituents are investigated via 1H-NMR metabolic technique.

The following points need to be carefully revised or addressed:

  1. Line 52, please use lowercase for the initial of “chromatography”.
  2. Line 55-56, the sentence is hard to read. Please revise it.
  3. How many observations and significant components are used for each PCA model? How much total variances can be explained by these PCA models? The authors need to clarify this for each PCA result.
  4. For the supervised analyses, if the observations were divided into two sets, i.e., a training set to build the statistical model and an external set for validation purpose? The authors need to give more details about the OPLS-DA analysis on lines 101, 176-177, and 229, or directly in the method section of line 325.
  5. Line 117, please give the full name of VIP here. I also recommend a brief description of the VIP variable selection method, because not all readers are familiar with this approach.
  6. Line 119, please delete the extra period.
  7. Line 195, what is the sensory detection threshold of galacturonic acid? Please clarify this in the sentence.
  8. Line 226, please check the number of constituents in the boxplot.
  9. Lines 299 and 301, add city name right before France.
  10. Line 317, add city name right before Germany.
  11. Check and unify reference style according to journal instructions. Some reference titles were capitalized (Lines 384 and 385), while some are not (389-390).

Reviewer 2 Report

The work is investigating the ability of 1H-NMR spectroscopy associated with chemometric tools to distinguish different winemaking parameters and processes (maturity, fining and enzymatic adding) and to explore the impact of these processes on wine composition. The paper is well written and provided a nice approach and results. However, some information should be included in order to understand and enhance the work quality.  

 In general, the results were visualized by using PCA and OPLS-DA score plots and loading plots. The PCA discussion is limited, and no information about the explained variances. Could you please ad the explained variances for all your graphs and in the discussion?  However, the loading plots were not explained at all. The scores plots should be explained using the loading plots (for both PCA and OPLS-DA), from those graphs in the discussion indicate which chemicals are responsible for the discrimination. 

The authors stated that ''The highlighted wine constituents were further controlled by Student's t-test, only those with p-value > 0.05 were conserved''. How can you explain you have used only non-significant constituents (p-value > 0.05)? Normally we select the significant ones and the p-value should be (p-value < 0.05)? Did you compare one by one so the Student's t-test is only to compare between two means and you have three groups (M1, M2,and M3; E1, E2 and E3)? Why did you not apply a Tukey's test the compare directly the three of them? The statistical study should be reworked in a multivariate compariason or please explain why not?

The OPLS-DA models were validated by cross-validation, which kind of cross-validation procedure did you apply? please add and explain why? How many latent variables did you used to construct the models? Please add in your disscussion all those information.

Which kind of data preprocessing did you use for PCA and OPLS-DA? Please add for each model and explain why? if any preprocessing was applied also please explain why? All those remarks should be explained within the discussion and detailed in order to understand the work. 

In general, why do the authors prefer to use the OPLS-DA and not PLS-DA? Did you construct a new OPLS-DA model based only on those selected chemicals (VIP and p-(corr))? Did the classification was enhanced and perfect instead of the models constructed in the all data?  

Reviewer 3 Report

My suggested corrections are as follows:

I. Cite additional recent review literature when referring to the use of 1H NMR for wine analysis somewhere between lines 38 and 42:

  1. Solovyev, P.A.; Fauhl-Hassek, C; Riedl, J; Esslinger, S; Bontempo, L; Camin, C., NMR spectroscopy in wine authentication: An official control perspective. Compr Rev Food Sci Food Saf. 2021, 20, 2040-2062. https://doi.org/10.1111/1541-4337.12700
  2. Sobolev, A. P.; Thomas, F.; Donarski, J.; Ingallina, C.; Circi, S., Cesare Marincola, F.; ... Mannina, L., Use of NMR applications to tackle future food fraud issues. Trends in Food Science & Technology, 2019, 91, 347–353. https://doi.org/10.1016/j.tifs.2019.07.035

  3. Tabago, M. K. A. G.; Calingacion, M. N.; Garcia, J., Recent advances in NMR-based metabolomics of alcoholic beverages. Food Chemistry: Molecular Sciences, 2021, 2, 100009. https://doi.org/10.1016/j.fochms.2020.100009

II. The scores and loadings plot from SIMCA in Figures 2 (between lines 93 and 94), 3 (between lines 168 and 169) and 4 (between lines 223 and 224) have low resolution and are highly pixelated (rasterized). At the same time, the boxplots in Figures 3 and 4 are perfect vector images. SIMCA allows exporting 2-dimensional plots as vector images (EMF format). I propose to make the score and loading plots in vector forms as well for better appearance. The other alternative may be to increase resolution of exported images. The same concerns Figure 1: the spectrum curve and scale rulers are rasterized, while numbers and peak designations are not.

III. Small technical corrections.

In the Section 2.1. (1H NMR analysis) I'd suggest the following corrections:

In Table 1 the following groups are not subscripted as they should be:

a. Valine 1.04 (d, 7.3, CH3), CH3; 

b. Isopentanol  3.61 (t, 6.7, CH2)

c. Choline 3.19 (s, 3CH3)

In the Section 3.4 (NMR analysis) I'd suggest the following corrections:

a. Add more information about water and ethanol suppression (line 302) with values of power used for each signal, e.g. "power level utilized for water suppression pulse was 44.74 dB (50 hZ window)". The exact numbers can be easily found in the dataset.

b. Line 306, correct "90° pulse calibration was calibrated" either as "90° pulse was calibrated" or "90° pulse calibration was carried out".

Reviewer 4 Report

In this study the authors provide a 1H-NMR metabolomic analysis to discriminate wines produced from grapes at different stages of maturation providing informative trends on the potential quality of the wine. They also evaluate the effect of two different enzymes and two clarifying agents.

The work is well done from an NMR and statistical point of view. Minor revisions are required.

- Describe in the materials and methods part how the assignment of the metabolites was performed and better describe how the quantification of the metabolites was carried out.

- In the supplementary insert the p-values and express the significance in figures 1S and 2S.

- arrange the images in Figure 2

- line 120 p> 0.05? is there a mistake?

- line 119 punctuation error.

- line 197, remove the. But, ....

- fix spelling errors

Round 2

Reviewer 2 Report

The paper was corrected as suggested.